# Stochastic Network Design in Bidirected Trees

**Xiaojian Wu**[1]     **Daniel Sheldon**[1,2]     **Shlomo Zilberstein**[1]

[1] School of Computer Science, University of Massachusetts Amherst
[2] Department of Computer Science, Mount Holyoke College

## Abstract

We investigate the problem of stochastic network design in bidirected trees. In this problem, an underlying phenomenon (e.g., a behavior, rumor, or disease) starts at multiple sources in a tree and spreads in both directions along its edges. Actions can be taken to increase the probability of propagation on edges, and the goal is to maximize the total amount of spread away from all sources. Our main result is a rounded dynamic programming approach that leads to a fully polynomial-time approximation scheme (FPTAS), that is, an algorithm that can find $(1-\epsilon)$-optimal solutions for any problem instance in time polynomial in the input size and $1/\epsilon$. Our algorithm outperforms competing approaches on a motivating problem from computational sustainability to remove barriers in river networks to restore the health of aquatic ecosystems.

## 1   Introduction

Many planning problems from diverse areas such as urban planning, social networks, and transportation can be cast as *stochastic network design*, where the goal is to take actions to enhance connectivity in a network with some stochastic element [1–8]. In this paper we consider a simple and widely applicable model where a stochastic network $G'$ is obtained by flipping an independent coin for each edge of a directed host graph $G = (V, E)$ to determine whether it is included in $G'$. The planner collects reward $r_{st}$ for each pair of vertices $s, t \in V$ that are connected by a directed path in $G'$. Actions are available to increase the probabilities of individual edges for some cost, and the goal is to maximize the total expected reward subject to a budget constraint.

Stochastic network design generalizes several existing problems related to spreading phenomena in networks, including the well known *influence maximization* problem. Specifically, the coin-flipping process captures the live-edge characterization of the Independent Cascade model [7], in which the presence of edge $(u, v)$ in $G'$ allows *influence* (e.g., behavior, disease, or some other spreading phenomenon) to propagate from $u$ to $v$. Influence maximization seeks a seed set $S$ of at most $k$ nodes to maximize the expected number of nodes reachable from $S$, which is easily modeled within our model by assigning appropriate rewards and actions. The framework also captures more complex problems with actions that increase edge probabilities—a setup that proved useful in various computational sustainability problems aimed to restore habitat or remove barriers in landscape networks to facilitate the spread and conserve a target species [4–6, 8].

The stochastic network design problem in its general form is intractable. It includes influence maximization as a special case and is thus NP-hard to approximate within a ratio of $1 - 1/e + \epsilon$ for any $\epsilon > 0$ [7], and it is #P-hard to compute the objective function under fixed probabilities [9, 10]. Unlike the influence maximization problem, which is a monotone submodular maximization problem and thus admits a greedy $(1 - 1/e)$-approximation algorithm, the general problem is not submodular [6]. Previous problems in this class were solved by a combination of techniques including the sample average approximation, mixed integer programming, dual decomposition, and primal-dual heuristics [6, 11–13], none of which provide both scalable running-time and optimality guarantees.

It is therefore of great interest to design efficient algorithms with provable approximation guarantees for restricted classes of stochastic network design. Wu, Sheldon, and Zilberstein [8] recently showed that the special case in which $G$ is a directed tree where influence flows away from the root (i.e., rewards are non-zero only for paths originating at the root) admits a fully polynomial-time approximation scheme (FPTAS). Their algorithm—*rounded dynamic programming* (RDP)—is based on recursion over rooted subtrees. Their work was motivated by the *upstream barrier removal problem* in river networks [5], in which migratory fish such as salmon swim upstream from the root (ocean) of a river network attempting to access upstream spawning habitat, but are blocked by barriers such as dams along the way. Actions are taken to remove or repair barriers and thus increase the probability fish can pass and therefore utilize a greater amount of their historical spawning habitat.

In this paper, we investigate the harder problem of stochastic network design in a *bidirected* tree, motivated by a novel conservation planning problem we term *bidirectional barrier removal*. The goal is to remove barriers to facilitate point-to-point movement in river networks. This applies to the much broader class of resident (non-migratory) fish species whose populations and gene-flow are threatened by dams and smaller river barriers (e.g., culverts) [14]. Replacing or retrofitting barriers with passage structures is a key conservation priority [15, 16]. However, stochastic network design in a bidirected tree is apparently much harder than in a directed tree. Since spread originates at all vertices instead of a designated root and edges may have different probabilities in each direction, it is not obvious how computations can be structured in a recursive fashion as in [8].

Our main contribution is a novel RDP algorithm for stochastic network design in bidirected trees and a proof that it is an FPTAS—in particular, it computes $(1 - \epsilon)$-optimal solutions in time $O(n^8/\epsilon^6)$. To derive the new RDP algorithm, we first show in Section 3 that the computation can be structured recursively despite the lack of a fixed orientation to the tree by choosing an arbitrary orientation and using a more nuanced dynamic programming algorithm. However, this algorithm does not run in polynomial time. In Section 4, we apply a rounding scheme and then prove in Section 5 that this leads to a polynomial-time algorithm with the desired optimality guarantee. However, the running time of $O(n^8/\epsilon^6)$ limits scalability in practice, so in Section 6 we describe an adaptive-rounding version of the algorithm that is much more efficient. Finally, we show that RDP significantly outperforms competing algorithms on the bidirectional barrier removal problem in real river networks.

## 2    Problem Definition

The input to the stochastic network design problem consists of a bidirected tree $\mathcal{T} = (V, E)$ with probabilities $p_{uv}$ assigned to each directed edge $(u, v) \in E$. A finite set of possible repair actions $A_{u,v} = A_{v,u}$ is associated with each *bidirected* edge $\{u, v\}$; action $a \in A_{u,v}$ has cost $c_{uv,a}$ and, if taken, simultaneously increases the two directed edge probabilities to $p_{uv|a}$ and $p_{vu|a}$. We assume that $A_{u,v}$ contains a default zero-cost "noop" action $a_0$ such that $p_{uv|a_0} = p_{uv}$ and $p_{vu|a_0} = p_{vu}$. A policy $\pi$ selects an action $\pi(u, v)$—either a repair action or a noop—for each bidirected edge. We write $p_{uv|\pi} := p_{uv|\pi(u,v)}$ for the probability of edge $(u, v)$ under policy $\pi$. In addition to the edge probabilities, a non-negative reward $r_{s,t}$ is specified for each pair of vertices $s, t \in V$.

Given a policy $\pi$, the *s-t accessibility* $p_{s \rightsquigarrow t|\pi}$ is the product of all edge probabilities on the unique path from $s$ to $t$, which is the probability that $s$ retains a path to $t$ in the subgraph $\mathcal{T}'$ where each edge is present independently with probability $p_{uv|\pi}$. The total expected reward for policy $\pi$ is $z(\pi) = \sum_{s,t \in V} r_{s,t} \, p_{s \rightsquigarrow t|\pi}$. Our goal is to find a policy that maximizes $z(\pi)$ subject to a budget $b$ limiting the total cost $c(\pi)$ of the actions being taken. Hence, the resulting policy satisfies $\pi^* \in \arg\max_{\{\pi | c(\pi) \leq b\}} z(\pi)$.

In this work, we will assume that the rewards factor as $r_{s,t} = h_s h_t$, which is useful for our dynamic programming approach and consistent with several widely used metrics. For example, *network resilience* [17] is defined as the expected number of node-pairs that can communicate after random component failures, which is captured in our framework by setting $r_{s,t} = h_s = h_t = 1$. Network resilience is a general model of connectivity that can apply in diverse complex network settings. The ecological measure of *probability of connectivity* (PC) [18], which was the original motivation of our formulation, can also be expressed using factored rewards. PC is widely used in ecology and conservation planning and is implemented in the Conefor software, which is the basis of many planning applications [19]. A precise definition of PC appears below.

**Barrier Removal Problem** Fig. 1 illustrates the bidirectional barrier removal problem in river networks and its mapping to stochastic network design in a bidirected tree. A river network is a tree with edges that represent stream segments and nodes that represent either stream junctions or barriers that divide segments. Fish begin in each segment and can swim freely between adjacent segments, but can only pass a barrier with a specified *passage probability* or *passability* in each direction; in most cases, downstream passability is higher than upstream passability. To map this problem to stochastic network design, we create a bidirected tree $\mathcal{T} = (V, E)$ where each node $v \in V$ represents a contiguous region of the river network—i.e., a connected set of stream segments among which fish can move freely without passing any barriers—and the value $h_v$ is equal to the total amount of habitat in that region (e.g., the total length of all segments). Each barrier then becomes a bidirected edge that connects two regions, with the passage probabilities in the upstream and downstream directions assigned to the corresponding directed edges. It is easy to see that $\mathcal{T}$ retains a tree structure.

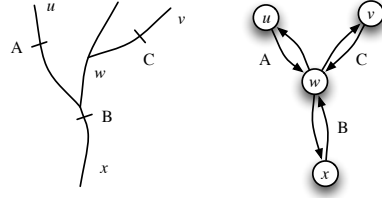

Figure 1: Left: sample river network with barriers A, B, C and contiguous regions $u, v, w, x$. Right: corresponding bidirected tree.

Our objective function $z(\pi)$ is motivated by PC introduced above. It is defined as follows:

$$PC(\pi) = \frac{z(\pi)}{R} = \frac{\sum_{s \in S} \sum_{t \in S} r_{s,t} p_{s \rightsquigarrow t | \pi}}{R} \tag{1}$$

where $R = \sum_{s,t} h_s h_t$ is a normalization constant. When $h_v$ is the amount of suitable habitat in region $v$, $PC(\pi)$ is the probability that a fish placed at a starting point chosen uniformly at random from suitable habitat (so that a point in region $s$ is chosen with probability proportional to $h_s$) can reach a random target point also chosen uniformly at random by passing each barrier in between.

In the rest of the paper, we present algorithms for solving this problem and their theoretical analysis that generalize the rounded DP approach introduced in [8].

# 3 Dynamic Programming Algorithm

Given a bidirected tree $\mathcal{T}$, we present a divide-and-conquer method to evaluate a policy $\pi$ and a dynamic programming algorithm to optimize the policy. We use the fact that given an arbitrary *root*, any bidirected tree $\mathcal{T}$ can be viewed as a rooted tree in which each vertex $u$ has corresponding *children* and *subtrees*. To simplify our algorithm and proofs, we make the following assumption.

**Assumption 1.** *Each vertex in the rooted tree has at most two children.*

Any problem instance can be converted into one that satisfies this assumption by replacing any vertex $u$ with more than two children by a sequence of internal vertices with exactly two children. The original edges are attached to the original children of $u$ and the added edges have probabilities 1. In the modified tree, $u$ has two children and its habitat is split equally among $u$ and the newly added vertices. The resulting binary tree has at most twice as many vertices as the original one. Most importantly, a policy for the modified tree can be trivially mapped to a unique policy for the original tree with the same expected reward.

**Evaluating A Fixed Policy Using Divide and Conquer** To evaluate a fixed policy $\pi$, we use a divide and conquer method that recursively computes a tuple of three values per subtree. Let $v$ and $w$ be the children of $u$. The tuple of the subtree $\mathcal{T}_u$ rooted at $u$ can be calculated using the tuples of subtrees $\mathcal{T}_v$ and $\mathcal{T}_w$. Once the tuple of $\mathcal{T}_{root} = \mathcal{T}$, is calculated, we can extract the total expected reward from that tuple.

Now, given a policy $\pi$, we define the tuple of $\mathcal{T}_u$ as $\psi_u(\pi) = (\nu_u(\pi), \mu_u(\pi), z_u(\pi))$, where

- $\nu_u(\pi) = \sum_{t \in \mathcal{T}_u} p_{u \rightsquigarrow t | \pi} h_t$ is the sum of the $s$-$t$ accessibilities of all paths from $u$ to $t \in \mathcal{T}_u$, each of which is weighted by the habitat $h_t$ of its ending vertex $t$.

- $\mu_u(\pi) = \sum_{s \in \mathcal{T}_u} p_{s \rightsquigarrow u | \pi} h_s$ is the sum of the $s$-$t$ accessibilities of all paths from $s \in \mathcal{T}_u$ to $u$, each of which is weighted by the habitat $h_s$ of its departing vertex $s$.

- $z_u(\pi) = \sum_{s \in \mathcal{T}_u} \sum_{t \in \mathcal{T}_u} p_{s \rightsquigarrow t | \pi} r_{s,t}$ ($r_{s,t} = h_s h_t$) represents the total expected reward that a fish obtains by following paths with both starting and ending vertices in $\mathcal{T}_u$.

The tuple $\psi_u(\pi)$ is calculated recursively using $\psi_v(\pi)$ and $\psi_w(\pi)$. To calculate $\nu_u(\pi)$, we note that a path from $u$ to a vertex in $\mathcal{T}_u \backslash \{u\}$ is the concatenation of either the edge $(u, v)$ with a path from $v$ to $\mathcal{T}_v$ or the edge $(u, w)$ with a path from $w$ to $\mathcal{T}_w$, that is, $\nu_u(\pi)$ can be written as

$$\sum_{t \in \mathcal{T}_v} p_{uv|\pi} p_{v \rightsquigarrow t|\pi} h_t + \sum_{t \in \mathcal{T}_w} p_{uw|\pi} p_{w \rightsquigarrow t|\pi} h_t + h_u = p_{uv|\pi} \nu_v(\pi) + p_{uw|\pi} \nu_w(\pi) + h_u \quad (2)$$

Similarly, $\mu_u(\pi) =$

$$\sum_{s \in \mathcal{T}_v} p_{s \rightsquigarrow v|\pi} p_{vu|\pi} h_s + \sum_{s \in \mathcal{T}_w} p_{s \rightsquigarrow w|\pi} p_{wu|\pi} h_s + h_u = p_{vu|\pi} \mu_v(\pi) + p_{wu|\pi} \mu_w(\pi) + h_u \quad (3)$$

By dividing the reward from paths that start and end in $\mathcal{T}_u$ based on their start and end nodes, we can express $z_u(\pi)$ as follows:

$$z_u(\pi) = z_v(\pi) + z_w(\pi) + \mu_v(\pi) p_{v \rightsquigarrow w|\pi} \nu_w(\pi) + \mu_w(\pi) p_{w \rightsquigarrow v|\pi} \nu_v(\pi) + h_u \nu_u(\pi) + h_u \mu_u(\pi) - h_u^2 \quad (4)$$

The first two terms describe paths that start and end within a single subtree—either $\mathcal{T}_v$ or $\mathcal{T}_w$. The third and fourth terms describe paths that start in $\mathcal{T}_v$ and end in $\mathcal{T}_w$ or vice versa. The last three terms describe paths that start or end at u, with an adjustment to avoid double-counting the trivial path that starts *and* ends at $u$. That way, all tuples can be evaluated with one pass from the leaves to the root and each vertex is only visited once. At the root, $z_{root}(\pi)$ is the expected reward of policy $\pi$.

**Dynamic Programming Algorithm**  We introduce a DP algorithm to compute the optimal policy. Let subpolicy $\pi_u$ be the part of the full policy that defines actions for barriers within $\mathcal{T}_u$. In the DP algorithm, each subtree $\mathcal{T}_u$ maintains a list of tuples $\psi$ that are reachable by some subpolicies and each tuple is associated with a least-cost subpolicy, that is, $\pi_u^* \in \arg\min_{\{\pi_u | \psi_u(\pi_u) = \psi\}} c(\pi_u)$.

Let $v$ and $w$ be two children of $u$. We recursively generate the list of reachable tuples and the associated least-cost subpolicies using the tuples of $v$ and $w$. To do this, for each $\psi_v$, $\psi_w$, we first extract the corresponding $\pi_v^*$ and $\pi_w^*$. Then, using these two least-cost subpolicies of the children, for each $a \in A_{uv}$ and $a' \in A_{uw}$, a new subpolicy $\pi_u$ is constructed for $\mathcal{T}_u$ with cost $c(\pi_u) = c_{uv,a} + c_{uw,a'} + c(\pi_v^*) + c(\pi_w^*)$. Using Eqs. (2), (3) and (4), the tuple $\psi_u(\pi_u)$ of $\pi_u$ is calculated. If $\psi_u(\pi_u)$ already exists in the list (i.e., $\psi_u(\pi_u)$ was created by some other previously constructed subpolicies), we update the associated subpolicy such that only the minimum cost subpolicy is kept. If not, we add this tuple $\psi_u(\pi_u)$ and subpolicy $\pi_u$ to the list.

To initialize the recurrence, the list of a leaf subtree contains only a single tuple $(h_u, h_u, h_u^2)$ associated with an empty subpolicy. Once the list of $\mathcal{T}_{root}$ is calculated, we scan the list to pick a pair $(\psi_{root}^*, \pi^*)$ such that $(\psi_{root}^*, \pi^*) \in \arg\max_{\{(\psi_{root}, \pi) | c(\pi) \leq b\}} z_{root}$ where $z_{root}$ is the third element of $\psi_{root}$. Finally, $\pi^*$ is the returned optimal policy and $z_{root}^*$ is the optimal expected reward.

## 4 Rounded Dynamic Programming

The DP algorithm is not a polynomial-time algorithm because the number of reachable tuples increases exponentially as we approach the root. In this section, we modify the DP algorithm into a FPTAS algorithm. The basic idea is to discretize the continuous space of $\psi_u$ at each vertex such that there only exists a polynomial number of different tuples. To do this, the three dimensions are discretized using granularity factors $K_u^\nu$, $K_u^\mu$ and $K_u^z$ respectively such that the space is divided into a finite number of cubes with volume $K_u^\nu \times K_u^\mu \times K_u^z$.

For any subpolicy $\pi_u$ of $u$ in the discretized space, there is a rounded tuple $\hat{\psi}_u(\pi_u) = (\hat{\nu}_u(\pi_u), \hat{\mu}_u(\pi_u), \hat{z}_u(\pi_u))$ to underestimate the true tuple $\psi_u(\pi_u)$ of $\pi_u$. To evaluate $\hat{\psi}_u(\pi_u)$, we use the same recurrences as (2), (3) and (4), but rounding each intermediate value into a value in the discretized space. The recurrences are as follow:

$$\hat{\nu}_u^{sum}(\pi_u) = p_{uv|\pi_u} \hat{\nu}_v(\pi_u) + p_{uw|\pi_u} \hat{\nu}_w(\pi_u) + h_u \quad \hat{\mu}_u^{sum}(\pi_u) = p_{vu|\pi_u} \hat{\mu}_v(\pi_u) + p_{wu|\pi_u} \hat{\mu}_w(\pi_u) + h_u$$

$$\hat{\nu}_u(\pi_u) = K_u^\nu \left\lfloor \frac{\hat{\nu}_u^{sum}(\pi_u)}{K_u^\nu} \right\rfloor \qquad\qquad \hat{\mu}_u(\pi_u) = K_u^\mu \left\lfloor \frac{\hat{\mu}_u^{sum}(\pi_u)}{K_u^\mu} \right\rfloor \qquad (5)$$

$$\hat{z}_u(\pi_u) = K_u^z \cdot \tag{6}$$

$$\left\lfloor \frac{\hat{z}_v(\pi_u) + \hat{z}_w(\pi_u) + \hat{\mu}_v(\pi_u)p_{v \rightsquigarrow w|\pi_u}\hat{\nu}_w(\pi_u) + \hat{\mu}_w(\pi_u)p_{w \rightsquigarrow v|\pi_u}\hat{\nu}_v(\pi_u) + h_u\hat{\mu}_u^{sum}(\pi_u) + h_u\hat{\nu}_u^{sum}(\pi_u) - h_u^2}{K_u^z} \right\rfloor$$

The modified algorithm—rounded dynamic programming (RDP)—is the same as the DP algorithm, except that it works in the discretized space. Specifically, each vertex maintains a list of reachable rounded tuples $\hat{\psi}_u$, each one associated with a least costly subpolicy achieving $\hat{\psi}_u$, that is, $\pi_u^* \in \arg\min_{\{\pi_u|\hat{\psi}_u(\pi_u)=\hat{\psi}_u\}} c(\pi_u)$. Similarly to our DP algorithm, we generate the list of reachable tuples for each vertex using its children's lists of tuples. The difference is that to calculate the rounded tuple of a new subpolicy we use recurrences (5) and (6) instead of (2), (3) and (4).

## 5 Theoretical Analysis

We now turn to the main theoretical result:

**Theorem 1.** *RDP is a FPTAS. Specifically, let $OPT$ be the value of the optimal policy. Then, RDP can compute a policy with value at least $(1 - \epsilon)OPT$ in time bounded by $O(\frac{n^8}{\epsilon^6})$.*

**Approximation Guarantee** Let $\pi^*$ be the optimal policy and let $\pi'$ be the policy returned by RDP. We bound the value loss $z(\pi^*) - z(\pi')$ by bounding the distance of the true tuple $\psi(\pi)$ and the rounded tuple $\hat{\psi}(\pi)$ for an arbitrary policy $\pi$. In Eqs. (5) and (6), starting from leaf vertices, each rounding operation introduces an error at most $K_u^{\cdot}$ where $\cdot$ represents $\nu$, $\mu$ and $z$.

For $\nu$, starting from $u$, each vertex $t \in \mathcal{T}_u$ introduces error $K_t^\nu$ by using the rounding operation. The error is discounted by the accessibility from $u$ to $t$. For $\mu$, each vertex $s \in \mathcal{T}_u$ introduces error $K_s^\mu$, discounted in the same way. The total error is equal to the sum of all discounted errors.

Finally, we get the following result by setting

$$K_u^\nu = \frac{\epsilon}{3}h_u, \quad K_u^\mu = \frac{\epsilon}{3}h_u, \quad K_u^z = \frac{\epsilon}{3}h_u^2 \tag{7}$$

**Lemma 1.** *If condition (7) holds, then for all $u \in V$ and an arbitrary policy $\pi$:*

$$\nu_u(\pi) - \hat{\nu}_u(\pi) \leq \sum_{t \in \mathcal{T}_u} p_{u \rightsquigarrow t|\pi}K_t^\nu = \frac{\epsilon}{3}\sum_{t \in \mathcal{T}_u} p_{u \rightsquigarrow t|\pi}h_t = \frac{\epsilon}{3}\nu_u(\pi) \tag{8}$$

$$\mu_u(\pi) - \hat{\mu}_u(\pi) \leq \sum_{s \in \mathcal{T}_u} p_{s \rightsquigarrow u|\pi}K_s^\mu = \frac{\epsilon}{3}\sum_{s \in \mathcal{T}_u} p_{s \rightsquigarrow u|\pi}h_s = \frac{\epsilon}{3}\mu_u(\pi) \tag{9}$$

The difference of $z(\pi) - \hat{z}(\pi)$ is bounded by the following lemma.

**Lemma 2.** *If condition (7) holds, $z(\pi) - \hat{z}(\pi) \leq \epsilon z(\pi)$ for an arbitrary policy $\pi$.*

The proof by induction on the tree appears in the supplementary material.

**Theorem 2.** *Let $\pi^*$ and $\pi'$ be the optimal policy and the policy return by RDP respectively. Then, if condition (7) holds, we have $z(\pi^*) - z(\pi') \leq \epsilon z(\pi^*)$.*

*Proof.* By Lemma 2, we have $z(\pi^*) - \hat{z}(\pi^*) \leq \epsilon z(\pi^*)$. Furthermore, $z(\pi') \geq \hat{z}(\pi') \geq \hat{z}(\pi^*)$ where the second inequality holds because $\pi'$ is the optimal policy with respect to the rounded policy value. Therefore, we have $z(\pi^*) - z(\pi') \leq z(\pi^*) - \hat{z}(\pi^*)$ which proves the theorem. $\square$

**Runtime Analysis** Now, we derive the runtime result of Theorem 1, that is, if condition (7) holds, the runtime of RDP is bounded by $O(\frac{n^8}{\epsilon^6})$. First, it is reasonable to make the following assumption:

**Assumption 2.** *The value $h_u$ is constant with respect to $n$ and $\epsilon$ for each $u \in V$.*

Let $m_{u,\hat{\nu}}$, $m_{u,\hat{\mu}}$ and $m_{u,\hat{z}}$ be the number of different values for $\hat{\nu}_u$, $\hat{\mu}_u$ and $\hat{z}_u$ respectively in the rounded value space of $u$.

**Lemma 3.** *If condition (7) holds, then*

$$m_{u,\hat{\nu}} = O\left(\frac{n_u}{\epsilon}\right), \qquad m_{u,\hat{\mu}} = O\left(\frac{n_u}{\epsilon}\right), \qquad m_{u,\hat{z}} = O\left(\frac{n_u^2}{\epsilon}\right) \tag{10}$$

*for all $u \in V$ where $n_u$ is the number of vertices in subtree $\mathcal{T}_u$.*

*Proof.* The number $m_{u,\hat{\nu}}$ is bounded by $\frac{\sum_{t\in\mathcal{T}_u} h_t}{K_u^\nu}$ where $\sum_{t\in\mathcal{T}_u} h_t$ is a naive and loose upper bound of $\nu_u$ obtained assuming all passabilities of streams in $\mathcal{T}_u$ are 1.0. By Assumption (2), $m_{u,\hat{\nu}} = O(\frac{n_u}{\epsilon})$. The upper bound of $m_{u,\hat{\mu}}$ can be similarly derived. Assuming all passabilities are 1.0, the upper bound of $z_u$ is $\sum_{s\in\mathcal{T}_u}\sum_{t\in\mathcal{T}_u} h_s h_t$. Therefore, $m_{u,\hat{z}} \leq \frac{\sum_{s\in\mathcal{T}_u}\sum_{t\in\mathcal{T}_u} h_s h_t}{K_u^z} = O(\frac{n_u^2}{\epsilon})$ □

Recall that RDP works by recursively calculating the list of reachable rounded tuples and associated least costly subpolicy. Using Lemma 3, we get the following main result:

**Theorem 3.** *If condition (7) holds, the runtime of RDP is bounded by $O(\frac{n^8}{\epsilon^6})$.*

*Proof.* Let $T(n)$ be the maximum runtime of RDP for any subtree with $n$ vertices. In RDP, for vertex $u$ with children $v$ and $w$, we compute the list and associated subpolicies by iterating over all combinations of $\hat{\psi}_v$ and $\hat{\psi}_w$. For each combination, we iterate over all available action combinations $a_{uv} \in A_{uv}$ and $a_{uw} \in A_{uw}$, which takes constant time because the number of available repair actions are constant w.r.t. $n$ and $\epsilon$. Therefore, we can bound $T(n)$ using the following recurrence:

$$T(n_u) = O(m_{v,\hat{\nu}} m_{v,\hat{\mu}} m_{v,\hat{z}} m_{w,\hat{\nu}} m_{w,\hat{\mu}} m_{w,\hat{z}}) + T(n_v) + T(n_w) \leq c\frac{n_v^4 n_w^4}{\epsilon^6} + T(n_v) + T(n_w)$$

$$\leq \max_{0 \leq k \leq (n_u-1)} c\frac{k^4(n_u - k - 1)^4}{\epsilon^6} + T(k) + T(n_u - k - 1)$$

where $n_u = 1 + n_v + n_w$ as $\mathcal{T}_u$ consists of $u$, $\mathcal{T}_v$ and $\mathcal{T}_w$. The second inequality is due to Lemma 3. The third inequality is obtained by a change of variable.

We prove that $T(n) \leq c\frac{n^8}{\epsilon^6}$ using induction. For the base case $n = 0$, we have $T(n) = 0$ and for the base case $n = 1$, the subtree only contains one vertex, so $T(n) = c$. Now assume that $T(k) \leq c\frac{k^8}{\epsilon^6}$ for all $k < n$. Then one can show that

$$T(n) \leq \max_{0 \leq k \leq (n-1)} \frac{c}{\epsilon^6}\left(k^4(n - k - 1)^4 + k^8 + (n - k - 1)^8\right) \leq c\frac{n^8}{\epsilon^6} \tag{11}$$

and thus the theorem holds. A detailed justification of the final inequality appears in the supplementary material. □

## 6 Algorithm Implementation and Experiments

The theoretical results suggest that the RDP approach may be impractical for large networks. However, we can accelerate the algorithm and produce high quality solutions by making some changes, motivated by observations from our initial experiments. First, the theoretical runtime upper bound is much worse than the actual runtime of RDP because in practice, because the number of reachable tuples per vertex is much lower than the upper bounds of $m_{u,\hat{\nu}}$ $m_{u,\hat{\mu}}$ and $m_{u,\hat{z}}$ used in the proof. Moreover, some inequalities used in Section 5 are very loose; most of the rounding operations in fact produce much less error than the upper bound $K_u^\cdot$. Therefore, we can set the values of $K_u^\cdot$ much larger than the theoretical values without compromising the quality of approximation.

Consequently, before calculating the list of reachable tuples of $u$, we first estimate the upper bound and lower bound of the reachable values of $\hat{\nu}_u$, $\hat{\mu}_u$ and $\hat{z}_u$ using the list of tuples of its children. Then, we dynamically assign values to $K_u^\cdot$ by fixing the total number of different discrete values of $\hat{\nu}_u$, $\hat{\mu}_u$ and $\hat{z}_u$ in the space, thereby determining the granularity of discretization. For example, if the upper and the lower bounds of $\hat{\nu}_u$ are 1000 and 500 respectively, and we want 10 different values, the value of $K_u^\nu$ is set to be $\frac{1000-500}{10} = 50$. By using a finer granularity of discretization, we get a slower algorithm but better solution quality. In our experiments, setting these numbers to be 50, 50 and 150 for $\hat{\nu}_u$, $\hat{\mu}_u$ and $\hat{z}_u$, the algorithm became very fast and we were able to get very good solution quality.

We compared RDP with a greedy algorithm and a state-of-the-art algorithm for conservation planning, which uses sample average approximation and mixed integer programming (SAA+MIP) [4, 6, 11]. We initially considered two different greedy algorithms. One incrementally maximizes the increase of expected reward. The other incrementally maximizes the ratio between increase in expected reward and action cost. We found that the former performs better than the latter, so we

only report results for that version. We compare all three algorithms on small river networks. On large networks, we only compare RDP with the greedy algorithm because SAA+MIP fails to solve problems of that size.

**Dataset** Our experiments use data from the CAPS project [20] for river networks in Massachusetts (Fig. 2). Barrier passabilities are calculated from barrier features using the model defined by the CAPS project. We created actions to model practical repair activities. For road-crossings, most passabilities start close to $1$ and are cheap to repair relative to dams. To model this, we set $A_{u,v} = \{a_1\}$, $p_{uv|a_1} = p_{vu|a_1} = 1.0$ and $c_{uv|a_1} = 5$. In contrast, it is difficult and expensive to remove dams, so multiple strategies must be considered to improve their passability. We created actions $A_u = \{a_1, a_2, a_3\}$ with action $a_1$ having $p_{uv|a_1} = p_{vu|a_1} = 0.2$ and $c_{uv|a_1} = 20$; action $a_2$ having $p_{uv|a_2} = p_{vu|a_2} = 0.5$ and $c_{uv|a_2} = 40$; and action $a_3$ having $p_{uv|a_3} = p_{vu|a_3} = 1.0$ and $c_{uv|a_3} = 100$.

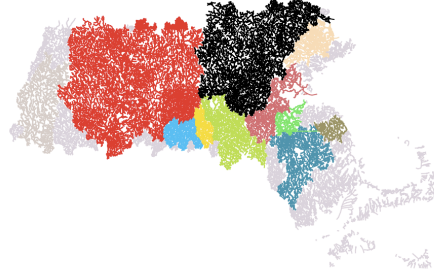

Figure 2: River networks in Massachusetts

**Results on Small Networks** We compared SAA+MIP, RDP and Greedy on small river networks. SAA+MIP used 20 samples for the sample average approximation and IBM CPLEX on 12 CPU cores to solve the integer program. RDP1 used finer discretization than RDP2, therefore requiring longer runtime. The results in Table 1 show that RDP1 gives the best increase in expected reward (relative to a zero-cost policy) in most cases and RDP2 produces similarly good solutions, but takes less time. Although Greedy is extremely fast, it produces poor solutions on some networks. SAA+MIP gives better results than Greedy, but fails to scale up. For example, on a network with 781 segments and 604 barriers, SAA+MIP needs more than 16G of memory to construct the MIP.

| Number of | | ER Increase | | | | Runtime | | | |
|---|---|---|---|---|---|---|---|---|---|
| Segments | Barriers | SAA+MIP | Greedy | RDP1 | RDP2 | SAA+MIP | Greedy | RDP1 | RDP2 |
| 106 | 36 | 3.7 | **4.1** | 4.1 | 4.0 | 3.3 | 0.0 | 0.7 | 0.4 |
| 101 | 71 | 4.0 | 3.6 | **4.3** | 4.3 | 19.5 | 0.0 | 2.5 | 1.2 |
| 163 | 91 | 11.3 | 11.2 | **12.3** | 12.1 | 42.3 | 0.0 | 13.6 | 6.8 |
| 263 | 289 | 20.7 | 11.1 | **25.3** | 24.8 | 1148.7 | 0.7 | 263.3 | 98.7 |
| 499 | 206 | 48.6 | **55.6** | 53.8 | 53.2 | 116.0 | 0.7 | 11.9 | 6.4 |
| 456 | 464 | 124.1 | 96.8 | **146.9** | 144.3 | 8393.5 | 0.7 | 359.9 | 142.0 |
| 639 | 609 | 51.8 | 25.8 | **53.7** | 51.6 | 12720.1 | 1.3 | 721.2 | 242.4 |

Table 1: Comparison of SAA, RDP and Greedy. Time is in seconds. Each unit of expected reward is $10^7$ (square meters). "ER increase" means the increase in expected reward after taking the computed policy.

**Results on Large Networks** We compared RDP and Greedy on a large network—the Connecticut River watershed, which has 10451 segments, 587 dams and 7545 crossings. We tested both algorithms on three different settings of action passabilities.

**Actions w/ symmetric passabilities** In this experiment, we used the actions introduced above. The expected reward increase (Fig. 3a) and runtime (Fig. 3b) are plotted for different budgets. For the expected reward, each unit represents $10^{14}\,\mathrm{m}^2$. Runtime is in seconds. As before, RDP1 uses finer discretization of tuple space than RDP2. As Fig. 3 shows, the RDP algorithms give much better so-

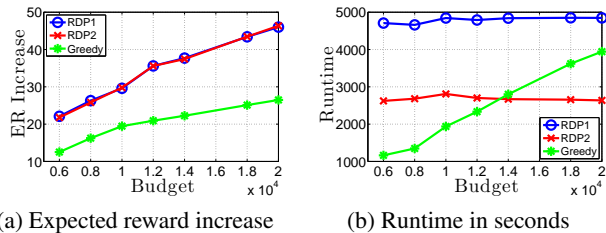

(a) Expected reward increase      (b) Runtime in seconds

Figure 3: RDP vs Greedy on symmetric passabilities.

lution quality than the greedy algorithm. With a budget of 20000, the ER increase of RDP1 is almost twice the increase for Greedy. Incidentally, RDP1 doesn't improve the solution quality by much, but it takes much longer time to finish. Notice that both RDP1 and RDP2 use constant runtime because the number of discrete values in both settings are bounded. In contrast, the runtime of Greedy increases with the budget size and eventually exceeds RDP2's runtime.

**Actions with asymmetric passabilities** The RDP algorithms work with asymmetric passabilities as well. For road-crossings, we set the actions to be the same as before. For dams, we first considered the case in which the downstream passabilities are all 1—which happens for some fish—and all upstream passabilities are the same as before. The results are shown in Figures 4a and 4b. In this case RDP still performs better than Greedy and tends to use less time as the budget increases.

We also considered a hard case in which the downstream passabilities of a dam are given by $p_{vu|a_1} = 0.8$, $p_{vu|a_2} = 0.9$, and $p_{vu|a_3} = 1.0$. These variations of passabilities produce more tuples in the discretized space. Our RDP algorithm still works well and produces better solutions than Greedy over a range of budgets as shown in Fig. 5a. As expected in such hard cases, RDP needs much

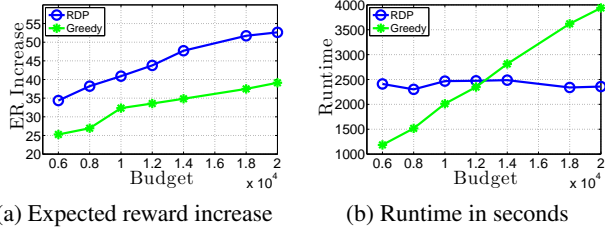

(a) Expected reward increase     (b) Runtime in seconds

Figure 4: RDP vs Greedy on asymmetric passabilities with all downstream passabilities equal to 1.

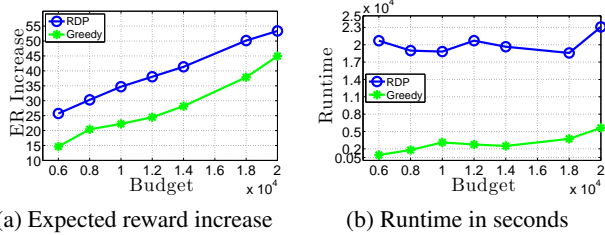

(a) Expected reward increase     (b) Runtime in seconds

Figure 5: RDP vs Greedy on asymmetric passabilities with varying downstream passabilities.

more time than Greedy. However, obtaining high quality solutions to such complex conservation planning problems in a matter of hours makes the approach very valuable.

**Time/Quailty Tradeoff** Finally, we tested the time/quality trade-off offered by RDP. The tradeoff is controlled by varying the level of discretization. We ran these experiments on the Connecticut River watershed using symmetric passabilities. Fig. 6 shows how runtime and expected reward grow as we refine the level of discretization. As we can see, in this case RDP converges quickly on high-quality results and exhibits the desired diminishing returns property of anytime algorithms—the quality gain is large initially and it diminishes as we continue to refine the discretization.

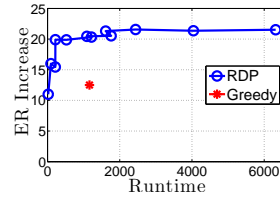

Figure 6: Time/quality tradeoffs

## 7 Conclusion

We present an approximate algorithm that extends the rounded dynamic programming paradigm to stochastic network design in bidirected trees. The resulting RDP algorithm is designed to maximize connectivity in a river network by solving the bidirectional barrier removal problem—a hard conservation planning problem for which no scalable algorithms exist. We prove that RDP is an FPTAS, returning $(1 - \epsilon)$-optimal solutions in polynomial time. However, its time complexity, $O(n^8/\epsilon^6)$, makes it hard to apply it to realistic river networks. We present an adaptive-rounding version of the algorithm that is much more efficient.

We apply this adaptive rounding method to segments of river networks in Massachusetts, including the entire Connecticut River watershed. In these experiments, RDP outperforms both a baseline greedy algorithm and an SAA+MIP algorithm, which is a state-of-art technique for stochastic network design. Our new algorithm offers an effective tool to guide ecologists in hard conservation planning tasks that help preserve biodiversity and mitigate the impacts of barriers in river networks. In future work, we will examine additional applications of RDP and ways to relax the assumption that the underlying network is tree-structured.

**Acknowledgments** This work has been partially supported by NSF grant IIS-1116917.

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
