[Supplementary Material · supplementary.pdf]

# Stochastic Network Design in Bidirected Trees
## — Supplementary Material —

**Xiaojian Wu**[1]     **Daniel Sheldon**[1,2]     **Shlomo Zilberstein**[1]

[1] School of Computer Science, University of Massachusetts Amherst
[2] Department of Computer Science, Mount Holyoke College

**Lemma 2.** *If condition (7) holds, $z(\pi) - \hat{z}(\pi) \leq \epsilon z(\pi)$ for an arbitrary policy $\pi$.*

*Proof.* To prove the lemma, we first use induction to prove the following statement:

$$z_u(\pi) - \hat{z}_u(\pi) \leq \epsilon z_u(\pi) \quad \forall u \in V$$

For the base case where $u$ is a leaf vertex, we have $z_u(\pi) = h_u^2$ and therefore the error $z_u(\pi) - \hat{z}_u(\pi)$ is bounded by $K_u^z = \frac{\epsilon}{3} z_u(\pi)$.

For the induction step, assume that the statement holds for the two children $v$ and $w$ of $u$. To prove that it also holds for $u$, we consider the error introduced by **each term** in recurrence (6) that uses the rounded values $\hat{\nu}_v(\pi)$ and $\hat{\mu}_w(\pi)$ instead of the true values $\nu_v(\pi)$ and $\mu_w(\pi)$. For the term $\hat{\mu}_v(\pi) p_{v \rightsquigarrow w | \pi} \hat{\nu}_w(\pi)$, the introduced error is $\left( \mu_v(\pi) p_{v \rightsquigarrow w | \pi} \nu_w(\pi) - \hat{\mu}_v(\pi) p_{v \rightsquigarrow w | \pi} \hat{\nu}_w(\pi) \right)$. Using Lemma 1, the error is

$$\leq \left( \frac{\epsilon}{3} \mu_v(\pi) \hat{\nu}_w(\pi) + \frac{\epsilon}{3} \hat{\mu}_v(\pi) \nu_w(\pi) + \frac{\epsilon^2}{3^2} \mu_v(\pi) \nu_w(\pi) \right) p_{v \rightsquigarrow w | \pi} \leq \epsilon \mu_v(\pi) p_{v \rightsquigarrow w | \pi} \nu_w(\pi)$$

where the last inequality holds because the rounded value always underestimates the true value.

Similarly, the error for the term $\hat{\mu}_w(\pi) p_{w \rightsquigarrow v | \pi} \hat{\nu}_v(\pi)$ is bounded by $\epsilon \mu_w(\pi) p_{w \rightsquigarrow v | \pi} \nu_v(\pi)$. For the term $h_u \hat{\mu}_u^{sum}$, the error is $h_u(\mu_u(\pi) - \hat{\mu}_u^{sum}(\pi))$. By using Lemma 1, the error is bounded by $h_u \frac{\epsilon}{3}(\mu_u(\pi) - h_u)$. Similarly, the error for the term $h_u \hat{\nu}_u^{sum}$ is bounded by $h_u \frac{\epsilon}{3}(\nu_u(\pi) - h_u)$. In addition, by the inductive assumption, the errors for $\hat{z}_v(\pi)$ and $\hat{z}_w(\pi)$ are bounded by $\epsilon z_v(\pi)$ and $\epsilon z_w(\pi)$ respectively.

Therefore, the total error for the enumerator of Equation (6) is bounded by

$$\epsilon(z_v(\pi) + z_w(\pi) + \mu_v(\pi) p_{v \rightsquigarrow w | \pi} \nu_w(\pi) + \mu_w(\pi) p_{w \rightsquigarrow v | \pi} \nu_v(\pi) + h_u \nu_{\cdot, u}(\pi) + h_u \mu_u(\pi) - 2 \cdot h_u^2)$$

According to the definition of $\hat{z}_u(\pi)$ in Equation (6), the enumerator is divided by $K_u^z$, rounded and then multiplied by $K_u^z$. These operations introduce an additional error $K_u^z$, which is bounded by $\epsilon h_u^2$ based on condition (7) in the paper. By adding the error to the total error above, the error $z_u(\pi) - \hat{z}_u(\pi)$ is bounded by $\epsilon z_u(\pi)$ where the expression of $z_u(\pi)$ is shown in Equation (4) in the paper. Hence, the statement holds for $u$.

Finally, as defined in the paper, $z(\pi) = z_{root}(\pi)$ and $\hat{z}(\pi) = \hat{z}_{root}(\pi)$. Therefore, the lemma holds. $\qquad\square$

**Derivation of the last inequality used to prove Theorem 3.**

$$\max_{0 \leq k \leq (n-1)} \frac{c}{\epsilon^6} \left( k^4(n-k-1)^4 + k^8 + (n-k-1)^8 \right) \leq c \frac{n^8}{\epsilon^6}$$

*Proof.* First, we show for any integer $k \in [0, n-1]$ that

$$
\begin{aligned}
& k^4(n-k-1)^4 + k^8 + (n-k-1)^8 \\
& \leq 2k^4(n-k-1)^4 + k^8 + (n-k-1)^8 \\
& = \left(k^4 + (n-k-1)^4\right)^2 \\
& \leq \left(k^4 + 2k^2(n-k-1)^2 + (n-k-1)^4\right)^2 = \left(\left(k^2 + (n-k-1)^2\right)^2\right)^2 \\
& \leq \left(\left(k^2 + 2k(n-k-1) + (n-k-1)^2\right)^2\right)^4 = \left((k+n-k-1)^2\right)^4 \\
& = (n-1)^8 \leq n^8
\end{aligned}
$$

Therefore, we have

$$
\max_{0 \leq k \leq (n-1)} k^4(n-k-1)^4 + k^8 + (n-k-1)^8 \leq n^8
$$

which proves the fact. $\qquad\square$