[Reviews · NeurIPS 2014]

Submitted by Assigned_Reviewer_29

This paper provides an FPTAS for stochastic network design in bidirected trees in time O(n^8/ε^6). The authors achieve this via Dynamic Programming.
Since this algorithm is pretty slow, they provide a more efficient algorithm and give empirical results on that.

I'm not sure about the relevance of the Stochastic Network Design Problem at NIPS, but given that it generalizes the Influence Maximization problem, there should be interest.

In the rebuttal I am happy to hear that this paper's algorithm is the best known in terms of worst-case complexity.
Summary: This paper provides an FPTAS for stochastic network design in bidirected trees in time O(n^8/ε^6).

Submitted by Assigned_Reviewer_50

The paper describes a generalization of an algorithm (stochastic network design) for the undirected case with multiple sources.
The proposed algorithm solves a real life problem (improving fish migration).
The approximation algorithm is polynomial, but his high (8) degree makes it problematic.
Summary: The algorithm is novel and expands on the prior art.
Though the high degree polynomial*, the authors managed to solve a real life problem.

* Thanks for the authors response - I now understand that while the 8th degree polynomial is theoretically important, in real life it runs much faster.

Submitted by Assigned_Reviewer_51

The paper generalize a previous work on stochastic network design on a directed rooted tree to model influence propagating from a single source to the whole network. In this paper, the directions on the edges are in both way and not directed away from the source as in the previous work. The paper generalizes the rounded dynamic programming approach from the previous work to get an FPTAS for the more general setting. The idea is to discretize the space of possible outcomes at any subtree. For each subtree rooted at a vertex v, there are 3 parameters defining the outcome: the influence within the subtree, the influence going through v to the subtree, and the influence going from the subtree through v to the rest of the tree. This crucially depends on the assumption that the reward factor/influence for 2 nodes s and t must be of the form h(s)*h(t) for a fixed function h. This assumption allows for aggregating influence going through v into 2 parameters as described above.

The writing is clear and easy to follow. The dynamic programming approach is standard in the literature for designing FPTAS and in this particular case, quite similar to previous work. The setup is more general than in previous work but there is still the restriction that the reward factor for 2 nodes s and t must be of the form h(s)*h(t) for a fixed function h. It is only mentioned that this restriction is a good match for the application without any further explanation.

== added after rebuttal ==
Thanks for adding applications where the reward factors of the form h(s)*h(t) make sense.
Summary: The paper generalize previous work on stochastic network design on a rooted tree to the setting where influence can propagate in both directions at different rates. The proposed algorithm is similar to previous work and follows standard approach in the literature.

Submitted by Assigned_Reviewer_52

The paper addresses a subclass of stochastic network design problem that has bi-directed tree structures. A rounded dynamic programming scheme is proposed to find (1-\epsilon)-optimal solutions in run time O(n^8 / \epsilon^6). Efficient implementation tricks are introduced to speed up the algorithm.

This is a well-written paper, but the downside is that this rounded DP algorithm can only work under the assumption that the rewards factor as r_{s,t}=h_s h_t, which might limit the possibility of generalizing this method to more generic applications.

One comment on the experiments:
It's mentioned in the paper (line 338) that the Greedy baseline is implemented as, in each iteration, choosing an edge that incrementally maximizes the increase of ER. I think, however, a better greedy implementation would be to take the cost of an edge into account, i.e., to choose the edge whose incremental gain normalized by its cost is the maximum. It might make a big difference when the costs vary significantly across edges. And in general, this normalized greedy implementation always outperforms the one used in the paper. Therefore, it would be interesting to empirically compare RDP to this normalized greedy implementation.

Post-rebuttal:
I am still curious to see how the normalized greedy performs relative to the method proposed in this work. But in general, I think this paper should be accepted.

Summary: A rounded Dynamic Programming (RDP) method is introduced to solve a subclass of stochastic network design problem that admits bi-directed tree-structure. Though running in high order polynomial time, efficient implementation tricks are introduced to speed up the algorithm.

This is a well-written paper, and it presents an interesting framework for an interesting real-life problem. But the greedy baseline that RDP is compared against might not be optimum. I am curious to see if the same empirical improvements can be carried over against the normalized greedy baseline.
Author Feedback
Author rebuttal: Thank you for the feedback and suggestions.

We wish to clarify a few issues raised in the reviews.

Regarding the complexity of the algorithm, our worst-case analysis is a conventional paradigm to assess complexity and, in fact, our algorithm is the best known in terms of this measure. More importantly, the algorithm performs well in practice -- better than the worst-case complexity (upper bound) indicates. Our experiments show that the approach is effective in solving the problem for the largest actual river network in Massachusetts.

Regarding the broader applicability and relevance of the approach, we note that the algorithm is generally applicable to a new class of influence maximization problems that go beyond source selection and examine costly actions that increase or decrease infection probabilities. This can be used for a wide range of applications such as suppressing the spread of fires or viruses or allocating limited funds to fixing roads and bridges after a natural disaster -- a problem that we address in current work.

Regarding the assumption that r_st = h_s * h_t, which is needed to derive the complexity result, we note that it is consistent with several widely used metrics. An example that we didn’t mention in the paper is the “network resilience” measure from the classical field of network reliability:

Network resilience, Charles J Colbourn - SIAM Journal on Algebraic Discrete Methods, 1987.

Network resilience is defined as the expected number of node-pairs that can communicate after random component failures, which is captured in our framework by setting r_st = h_s = h_t = 1. This is a general and very natural model of connectivity that one can imagine applying in diverse complex network settings. We will add this reference to the paper as further motivation.

The “probability of connectivity” (PC) measure was our original motivation and is the natural generalization of network resilience to the case where nodes represent areas of different sizes. PC is widely used in ecology and conservation planning. The original paper has about 200 citations, and PC is implemented in the Conefor software, which is the basis of many applications:

http://www.conefor.org/applications.html

We consider it an interesting but challenging line of future work to extend provable approximation guarantees to more general reward and graph structures.

With respect to prior work on directed trees [1], we note that the dynamic programming algorithm in this paper is entirely different. We have a richer parameter representation of each table entry, making the FPTAS proof a nontrivial generalization. Additionally, we employ sparse matrix representation methods that contribute significantly to the overall performance.